# Relationship between the Molecular Geometry and the Radiative Efficiency in Naphthyl-Based Bis-Ortho-Carboranyl Luminophores

**DOI:** 10.3390/molecules27196565

**Published:** 2022-10-04

**Authors:** Sanghee Yi, Mingi Kim, Chan Hee Ryu, Dong Kyun You, Yung Ju Seo, Kang Mun Lee

**Affiliations:** Department of Chemistry, Institute for Molecular Science and Fusion Technology, Kangwon National University, Chuncheon 24341, Korea

**Keywords:** *closo*-*ortho*-carborane, naphthalene, intramolecular charge transfer, orthogonality, radiative decay

## Abstract

The efficiency of intramolecular charge transfer (ICT)-based emission on π-aromatic-group-appended *closo-ortho*-carboranyl luminophores is known to be affected by structural fluctuations and molecular geometry, but investigation of this relationship has been in progress to date. In this study, four naphthyl-based bis-*o*-carboranyl compounds, in which hydrogen (**15CH** and **26CH**) or trimethysilyl groups (**15CS** and **26CS**) were appended at the *o*-carborane cage, were synthesized and fully characterized. All the compounds barely displayed an emissive trace in solution at 298 K; however, **15CH** and **26CH** distinctly exhibited a dual emissive pattern in rigid states (in solution at 77 K and in films), attributed to locally excited (LE) and ICT-based emission, while **15CS** and **26CS** showed strong ICT-based greenish emission. Intriguingly, the molecular structures of the four compounds, analyzed by single X-ray crystallography, showed that the C-C bond axis of the *o*-carborane cage in the trimethysilyl group-appended compounds **15CS** and **26CS** were more orthogonal to the plane of the appended naphthyl group than those in **15CH** and **26CH**. These features indicate that **15CS** and **26CS** present an efficient ICT transition based on strong *exo*-π-interaction, resulting in a higher quantum efficiency (Φ_em_) for ICT-based radiative decay than those of **15CH** and **26CH**. Moreover, the **26CS** structure revealed most orthogonal geometry, resulting in the highest Φ_em_ and lowest *k*_nr_ values for the ICT-based emission. Consequently, all the findings verified that efficient ICT-based radiative decay of aromatic group-appended *o*-carboranyl luminophores could be achieved by the formation of a specific geometry between the *o*-carborane cage and the aromatic plane.

## 1. Introduction

Naphthalene-based organic compounds have been widely used as components of optoelectronic materials in organic light-emitting diodes [1,2,3], photovoltaic cells [4,5,6,7,8,9,10,11], and organic thin-film transistors [12,13,14,15,16] because of their outstanding electrochemical and chemical stability and outstanding photophysical characteristics. These unique properties are mostly attributed to the electronic-abundant nature and structural rigidity originating from the dimeric fused-benzene ring formation [17]. In particular, these optical features can be manipulated by the electronic characteristics of the introduced functional groups, leading to the extensive utilization of naphthyl derivatives in various industrial fields [18,19,20,21,22,23,24]. Among the specific functional units showing consistent electronic tendency and chemical stability, the *ortho*-carborane of *closo*-type structure, a well-known icosahedral boron cluster recognized as a three-dimensional subspecies of the benzene ring [25,26,27,28], has recently become a topic of interest for inducing the revelation of intriguing photophysical characteristics [29,30,31]. The *o*-carborane composed of ten boron atoms intrinsically has a strong electron-withdrawing nature via C-substituents and the high polarizability of its σ-aromatic framework, resulting in intramolecular charge-transfer (ICT) transitions during photoexcitation and relaxation processes in clusters bearing electron-rich aromatic groups [30,31,32,33,34,35,36,37,38,39,40,41]. As a result, the compound in which the π-conjugated aryl group is linked to the carbon atom of the *o*-carborane can be called a donor (D, aromatic group)–acceptor (A, *o*-carborane) dyad system. The naphthyl moiety is a candidate for several π-aromatic scaffolds in the *o*-carboranyl dyad compounds, exhibiting interesting emissive features based on the ICT transition [40,42,43,44,45]. Chujo et al. reported emissive color-tuning properties from the blue to near-infrared energy region using bis-*o*-carborane-substituted naphthalene derivatives [46].

The luminescent characteristics of *o*-carboranyl dyad compounds can be significantly affected by the molecular geometry between the appended aromatic group and *o*-carborane cage and can furthermore strongly control the revelation and severance of ICT electronic transition [40,41,47,48,49]. For example, Fox et al. reported *C*-diazaboryl-*o*-carborane dyads demonstrating dual emissions from high-energy locally excited (LE) and low-energy CT states depending on the rotational motion of the diazaboryl substituents [50]. Our group also revealed that *o*-caboranes bearing a spirobifluorene moiety fluorescent donor could exhibit multiple photoluminescent characteristics originating from the alternation of the ICT process via inhibition of twisted fluctuations around the *o*-carborane cage, resulting in the dyad compounds being utilized as a visual sensory material [51]. These previous works distinctly indicate that the relationship between the geometric formation of aromatic rings and the *o*-carborane cage is a critical factor in adjusting the efficiency of the ICT-based radiative decay mechanism in *o*-carboranyl dyad luminophores. Although various aromatic group-appended D–A-type *o*-carboranyl luminophores revealed the changed photophysical features triggered by the twisted ICT phenomenon [52,53,54,55,56,57,58], extensive investigation of this specific molecular geometry, which can reinforce the ICT transition and enhance the ICT-based radiative decay efficiency, is still in progress.

In line with the research trends and intrinsic concept of the photophysical properties of *o*-carboranyl luminophores, four types of *o*-carboranyl D–A-type conjugated compounds based on a popular aromatic fluorophore, the naphthalene group, were investigated. We strategically designed naphthyl-bis-*o*-carboranyl compounds to confirm a geometric key affecting their photophysical properties and electronic transitions; *o*-carborane cages were introduced to the 1,5- or 2,6-position of naphthalene and hydrogen atoms (the smallest functional group, **15CH** and **26CH**, Figure 1) and trimethylsilyl groups (bulky group, **15CS** and **26CS**) were substituted with the carbon atoms on the *o*-carboranes. A comparison of the molecular structures and photophysical analysis of each naphthyl-based *o*-carborane compound enabled us to elucidate whether the specific molecular geometry is crucial to the efficiency of the ICT-based radiative decay mechanism. The detailed synthetic procedures, full characterization including molecular structure analysis, and photophysical properties (with theoretical calculations) are described below.

## 2. Materials and Methods

### 2.1. General Considerations

All operations were performed under an inert nitrogen atmosphere using standard Schlenk and glove box techniques. Anhydrous solvents (tetrahydrofuran (THF), triethylamine (NEt_3_), and toluene; Sigma-Aldrich, St. Louis, MO, USA) were dried by passing through an activated alumina column and were then stored over activated molecular sieves (5 Å). Spectrophotometric-grade solvents (THF, methanol (MeOH), ethyl acetate, *n*-hexane, and dichloromethane (DCM)) were used as received from Alfa Aesar (Haverhill, MA, USA). All commercial reagents were used without any further purification after purchase from Sigma-Aldrich (1,5-dibromonaphthalene, 2,6-dibromonaphthalene, ethynyltrimethylsilane, bis(triphenylphosphine)palladium(II) dichloride (Pd(PPh_3_)_2_Cl_2_), copper(I) iodide (CuI), 1.6 M *n*-butyllithium in hexane (*n*-BuLi), ammonium chloride (NH_4_Cl), magnesium sulfate (MgSO_4_), potassium carbonate (K_2_CO_3_), *N*,*N*-dimethylaniline, trimethylsilyl chloride (TMSCl), and poly(methyl methacrylate) (PMMA)), and Alfa Aesar (decaborane (B_10_H_14_), and basic alumina). Deuterated solvents (chloroform (CDCl_3_), dichloromethane (CD_2_Cl_2_), and THF-*d*_8_) were purchased from Cambridge Isotope Laboratories (Tewksbury, MA, USA) and dried over activated molecular sieves (5 Å). All nuclear magnetic resonance (NMR) spectra were recorded on a Bruker Avance 400 spectrometer (400.13 MHz for ^1^H and ^1^H{^11^B}, 100.62 MHz for ^13^C, and 128.38 MHz for ^11^B{^1^H}) at ambient temperature (Bruker Corporation, Billerica, MA, USA). Chemical shifts are given in ppm and are referenced to external tetramethylsilane (Me_4_Si) (^1^H, {^11^B}^1^H, and ^13^C) or BF_3_·Et_2_O ({^1^H}^11^B).

### 2.2. Synthesis of 1,5-Diethynylnaphthalene, 15AH

THF (14 mL) and NEt_3_ (10 mL) were added via cannula to a mixture of 1,5-dibromonaphthalene (0.60 g, 2.1 mmol), CuI (33 mg, 0.17 mmol), and Pd(PPh_3_)_2_Cl_2_ (0.12 g, 0.17 mmol) at ambient temperature. After stirring for 10 min, ethynyltrimethylsilane (0.64 mL, 4.6 mmol) was slowly added. The reaction mixture was then heated under reflux at 80 °C for 24 h prior to cooling to ambient temperature and evaporation of the solvent to afford a dark brown residue. Filtration through celite was employed to remove the metal salts from the reaction mixtures. The products were then used in situ for the next step, without characterization. More specifically, a round-bottom flask was charged with a mixture of the crude product (1,5-bis((trimethylsilyl)ethynyl)naphthalene, 0.58 g, 1.8 mmol) and K_2_CO_3_ (1.0 g, 7.2 mmol). Methanol (15 mL) was then added, and the mixture was stirred at 25 °C for 2 h. After this time, the reaction mixture was quenched by the addition of brine, and the aqueous phase was subsequently extracted with ethyl acetate (3 × 30 mL). The combined organic layer was dried over anhydrous MgSO_4_, filtered, and the volatiles were removed under vacuum to obtain a brown residue. Purification by column chromatography on silica (eluent: *n*-hexane) yielded **15AH**. Yield = 56% (0.21 g). ^1^H NMR (CDCl_3_): δ 8.40 (d, *J* = 8.0 Hz, 2H), 7.78 (d, *J* = 7.2, 2H), 7.53 (t, *J* = 8.4 Hz, 2H), 3.49 (s, 2H, acetylene-*H*). ^13^C NMR (CDCl_3_): δ 133.26, 131.83, 127.38, 126.14, 120.28, 82.44 (acetylene-*C*), 81.46 (acetylene-*C*). Anal. Calcd for C_14_H_8_: C, 95.42; H, 4.58. Found: C, 94.89; H, 4.23.

### 2.3. Synthesis of 2,6-Diethynylnaphthalene, 26AH

A procedure analogous to that for **15AH** was employed utilizing 2,6-dibromonaphthalene (1.4 g, 5.0 mmol), CuI (76 mg, 0.40 mmol), Pd(PPh_3_)_2_Cl_2_ (0.28 g, 0.40 mmol), and ethynyltrimethylsilane (1.5 mL, 11 mmol). Subsequently, the reaction between the crude 2,6-bis((trimethylsilyl)ethynyl)naphthalene (1.4 g, 4.2 mmol) and K_2_CO_3_ (2.3 g, 17 mmol) afforded the crude **26AH**, which was purified by silica gel column chromatography (eluent: *n*-hexane). Yield = 42% (0.37 g). ^1^H NMR (CDCl_3_): δ 7.99 (s, 2H), 7.75 (d, *J* = 8.4 Hz, 2H), 7.54 (d, *J* = 8.4 Hz, 2H), 3.18 (s, 2H, acetylene-*H*). ^13^C NMR (CDCl_3_): δ 132.41, 132.06, 129.36, 127.89, 120.49, 83.68(acetylene-*C*), 78.23(acetylene-*C*). Anal. Calcd for C_14_H_8_: C, 95.42; H, 4.58. Found: C, 94.95; H, 4.25.

### 2.4. Synthesis of 15CH

To a toluene solution (30 mL) of B_10_H_14_ (1.4 g, 11.7 mmol) and **15AH** (0.69 g, 3.9 mmol), an excess of *N*,*N*-dimethylaniline (2.5 mL, 20 mmol) was added at ambient temperature. After heating to reflux at 110 °C, the reaction mixture was stirred for 18 h. After this time, the volatiles were removed under vacuum and the resulting solid was dissolved in toluene. The solution was purified by column chromatography on basic alumina (eluent: toluene) to produce **15CH** as a white solid. Yield = 53% (0.85 g). ^1^H{^11^B} NMR (THF-*d*_8_): δ 9.08 (d, *J* = 8.9 Hz, 2H), 7.95 (d, *J* = 7.7 Hz, 2H), 7.59 (t, *J* = 8.3 Hz, 2H), 5.65 (s, 2H, CB-*H*), 2.94 (br s, 4H, CB-B*H*), 2.53 (br s, 6H, CB-B*H*), 2.34 (br s, 5H, CB-B*H*), 2.27 (br s, 5H, CB-B*H*). ^13^C NMR (THF-*d*_8_): δ 132.12, 130.98, 128.55, 128.49, 125.74, 78.43(CB-*C*), 64.04(CB-*C*). ^11^B{^1^H} NMR (THF-*d*_8_): δ −3.61 (br s, 6B, CB-*B*H), −9.62 (br s, 6B, CB-*B*H),−10.22 (br s, 4B, CB-*B*H), −14.09 (br s, 4B, CB-*B*H). Anal. Calcd for C_14_H_28_B_20_: C, 40.76; H, 6.84. Found: C, 40.55; H, 6.65.

### 2.5. Synthesis of 26CH

A procedure analogous to that for **15CH** was employed utilizing B_10_H_14_ (0.46 g, 3.8 mmol) and **26AH** (0.26 g, 1.5 mmol), and *N*,*N*-dimethylaniline (0.95 mL, 7.5 mmol). Purification by column chromatography on basic alumina (eluent: toluene) afforded **26CH** as a white solid. Yield = 73% (0.45 g). ^1^H{^11^B} NMR (CD_2_Cl_2_): δ 8.00 (s, 2H), 7.86 (d, *J* = 8.7 Hz, 2H), 7.63 (d, *J* = 8.8 Hz, 2H), 4.19 (s, 2H, CB-*H*), 2.66 (br s, 4H, CB-B*H*), 2.57 (br s, 4H, CB-B*H*), 2.42 (br s, 2H, CB-B*H*), 2.31 (br s, 10H, CB-B*H*). ^13^C NMR (CD_2_Cl_2_): δ 132.89, 132.85, 129.67, 127.58, 126.21, 76.71 (CB-*C*), 60.94 (CB-*C*).^11^B{^1^H} NMR (CD_2_Cl_2_): δ −3.14 (br s, 3B, CB-*B*H), −5.18 (br s, 3B, CB-*B*H), −9.88 (br s, 4B, CB-*B*H), −11.73 (br s, 6B, CB-*B*H), −13.65 (br s, 4B, CB-*B*H). Anal. Calcd for C_14_H_28_B_20_: C, 40.76; H, 6.84. Found: C, 40.60; H, 6.71.

### 2.6. Synthesis of 15CS

An *n*-hexane solution of *n*-BuLi (1.6 M, 1.6 mL, 2.5 mmol) was added dropwise to a solution of **15CH** (0.41 g, 1.0 mmol) in THF (10 mL) at 0 °C. After stirring for 1 h, the reaction was slowly allowed to warm to ambient temperature, and TMSCl (0.63 mL, 5.0 mmol) was added dropwise to the mixture. The reaction mixture was then stirred for 2 h at 25 °C. After quenching with saturated aqueous NH_4_Cl (15 mL), the mixture was extracted with ethyl acetate (3 × 15 mL). The combined organic layer was then dried over anhydrous MgSO_4_, filtered, and the solvent was removed under vacuum. The product was purified by recrystallization from DCM to obtain **15CS** as a white solid. Yield = 61% (0.34g). ^1^H NMR (CD_2_Cl_2_): δ 9.25 (d, *J* = 9.0 Hz, 2H), 8.32 (d, *J* = 7.7 Hz, 2H), 7.57 (t, *J* = 9.0 Hz, 2H), 3.30 (br s, 4H, CB-B*H*), 2.67 (br s, 6H, CB-B*H*), 2.48 (br s, 8H, CB-B*H*), 2.21 (br s, 2H, CB-B*H*), −0.31 (s, 18H, –Si(C*H*_3_)_3_). ^13^C NMR (CD_2_Cl_2_): δ 135.85, 133.30, 129.42, 129.22, 125.54, 85.99 (CB-*C*), 81.55 (CB-*C*), −0.29 (–Si(*C*H_3_)_3_). ^11^B{^1^H} NMR (CD_2_Cl_2_): δ −0.41 (br s, 3B, CB-*B*H), −2.06 (br s, 3B, CB-*B*H), −8.48 (br s, 6B, CB-*B*H), −11.00 (br s, 8B, CB-*B*H). Anal. Calcd for C_20_H_44_B_20_Si_2_: C, 43.13; H, 7.96. Found: C, 42.98; H, 7.80.

### 2.7. Synthesis of 26CS

A procedure analogous to that for **15CS** was employed utilizing *n*-BuLi (1.6 M, 0.78 mL, 1.3 mmol), **26CH** (0.20 g, 0.50 mmol), and TMSCl (0.32 mL, 2.5 mmol). The product was purified by recrystallization from *n*-hexane to obtain **26CS** as a white solid. Yield = 44% (0.12 g). ^1^H{^11^B} NMR (CD_2_Cl_2_): δ 8.18 (s, 2H), 7.88 (d, *J* = 8.4 Hz, 2H), 7.80 (d, *J* = 7.4 Hz, 2H), 3.04 (br s, 4H, CB-B*H*), 2.38 (br s, 12H, CB-B*H*), 2.26 (br s, 4H, CB-B*H*), −0.13 (s, 18H, –Si(C*H*_3_)_3_). ^13^C NMR (CD_2_Cl_2_): δ 133.02, 132.39, 131.53, 129.54, 129.52, 83.24 (CB-*C*), 77.63 (CB-*C*), −0.27 (–Si(*C*H_3_)_3_). ^11^B{^1^H} NMR (CD_2_Cl_2_): δ −0.23 (br s, 4B, CB-*B*H), −3.33 (br s, 4B, CB-*B*H), −8.89 (br s, 6B, CB-*B*H), −10.84 (br s, 3B, CB-*B*H), −12.53 (br s, 3B, CB-*B*H). Anal. Calcd for C_20_H_44_B_20_Si_2_: C, 43.13; H, 7.96. Found: C, 42.89; H, 7.77.

### 2.8. UV/Vis Absorption and Photoluminescence (PL) Measurements

Solution-phase UV/Vis absorption and PL measurements for each *o*-carborane compound were performed in degassed THF using a 1 cm quartz cuvette (50 µM) at 298 K. PL measurements were also carried out in THF at 77 K and in the film state (5 wt% doped in PMMA on a 15 × 15 mm quartz plate (thickness = 1 mm)). The UV/vis absorption and PL spectra were recorded on Jasco V-530 (Jasco, Easton, MD, USA) and FluoroMax-4P spectrophotometers (HORIBA, Edison, NJ, USA), respectively. The absolute PL quantum yields (Φ_em_) of the film samples were obtained using an absolute PL quantum yield spectrophotometer (FM-SPHERE, 3.2-inch internal integrating sphere on FluoroMax-4P, HORIBA, Edison, NJ, US) at 298 K. The fluorescence decay lifetimes of the films were measured at 298 K using a time-correlated single-photon counting (TCSPC) spectrometer (FLS920, Edinburgh Instruments, Livingston, UK) at the Central Laboratory of Kangwon National University. The TCSPC spectrometer was equipped with a pulsed semiconductor diode laser excitation source (EPL, 375 nm) and microchannel plate photomultiplier tube (MCP-PMT, 200–850 nm) detector.

### 2.9. X-ray Crystallography

Single-X-ray quality crystals of **15CH**, **26CH**, **15CS**, and **26CS** were grown from a DCM/*n*-hexane mixture. Single crystals were coated with paratone oil and mounted on a glass capillary. Crystallographic measurements were performed using a Bruker D8QUEST diffractometer (Bruker Coorperation, Billerica, MA, US), with graphite-monochromated Mo-Kα radiation (λ = 0.71073 Å) and CCD area detector. The structures of **15CH**, **26CH**, **15CS**, and **26CS** were assessed using direct methods, and all nonhydrogen atoms were subjected to anisotropic refinement with a full-matrix least-squares method on *F*^2^ using the SHELXTL/PC software package (released and presented at the Software Fayre, XXII IUCr Congress in Madrid, Spain, 2011). The X-ray crystallographic data are available in CIF format (CCDC 2184401–2184404 for **15CH**, **26CH**, **15CS**, and **26CS**), provided free of charge by the Cambridge Crystallographic Data Centre. The hydrogen atoms were placed at their geometrically calculated positions and refined using a riding model on the corresponding carbon atoms with isotropic thermal parameters. Detailed crystallographic data are provided in Appendix A.

### 2.10. Computational Calculation Studies

The optimized geometries for the ground (S_0_) and first excited (S_1_) states of all *o*-carboranyl compounds in THF were obtained at the B3LYP/6-31G(d,p) [59] level of theory. The vertical excitation energies at the optimized S_0_ geometries and the optimized geometries of the S_1_ states were calculated using time-dependent density functional theory (TD-DFT) [60] at the same level of theory. Solvent effects were evaluated using the conductor-like polarizable continuum model (CPCM) based on the integral equation formalism of the polarizable continuum model (IEFPCM), with THF as the solvent [61]. All geometry optimizations were performed using the Gaussian 16 program [62]. The percent contribution of a group in a molecule to each molecular orbital was calculated using the GaussSum 3.0 program [63]. The most stable geometries were determined by constructing one-dimensional potential energy surfaces as a function of each dihedral angle (Ψ, Appendix A) by rotating the carboranes of **15CH** and **26CH** between approximately 0° and 180° at intervals of 15° to yield 13 initial conformations for each compound. Conformations that exhibited physically impossible atomic overlaps were excluded from further geometric optimization. The dihedral angle was fixed, whereas other geometric variables were fully relaxed for geometry optimization and energy calculation of the resulting initial conformations using the Gaussian 16 software program [62]. The calculation method in this study does not provide accurate information for electronic transition of the *o*-carboranyl compounds and only gives an indirect hint to elucidate their transitions corresponding to specific photophysical characteristics.

## 3. Results and Discussion

### 3.1. Synthesis and Characterization

Naphthyl-bis-*o*-carboranyl compounds **15CH**, **26CH**, **15CS**, and **26CS**, in which the *o*-carborane cages are appended at the C1 and C5-positions of the C2 and C6-positions of the naphthyl group, were synthesized as shown in Figure 1. Sonogashira coupling reactions between ethynyltrimethylsilane and dibromonaphthanlene precursors (1,5-dibromonaphthalene for **15AH** and 2,6-dibromonaphthalene for **26AH**) and consecutive reactions with a weak base (K_2_CO_3_) for the deprotection of trimethylsilyl groups produced bis-ethynylnaphthalene compounds (**15AH** and **26AH**) in moderate yields (56% and 42%, respectively). The 1,5-or 2,6-bis [1-*o*-carboran-1-yl] naphthalene compounds (**15CH** and **26CH**) were prepared via cage-forming reactions with B_10_H_14_ using **15AH** or **26AH** in the presence of *N*,*N*-dimethylaniline (yields of 53% and 73%, respectively) [64,65,66]. Furthermore, lithiation of **15CH** and **26CH** with *n*-butyllithium followed by reaction with trimethylsilyl chloride afforded 1,5 or 2,6-bis [2-(trimethylsilyl)-1-*o*-carboran-1-yl]naphthanlene compounds (**15CS** and **26CS**) in 61% and 44% yields, respectively. The produced *o*-carboranyl compounds were stable in air and light and decomposed at detectable levels for more than six months under ambient conditions. All the precursors and prepared *o*-carboranyl naphthalene compounds (**15CH**, **26CH**, **15CS**, and **26CS**) were fully characterized using multinuclear (^1^H, ^1^H{^11^B}, ^13^C, and ^11^B{^1^H}) NMR spectroscopy (Appendix A) and elemental analysis. Each of the ^1^H{^11^B} NMR spectrum of four *o*-carboranyl compounds exhibited broad singlet peaks in the region of 3.3–2.2 ppm (totally integrated to 20 H atoms), confirming the existence of –BH units in the *closo*-*o*-carborane cages. In particular, a sharp signal around 0.0 ppm were observed in the ^1^H{^11^B} NMR spectra for **15CS** and **26CS**, which was attributed to the trimethylsilyl groups appended to *o*-carboranes. In addition, several broad singlet peaks were observed between −0 and −15 ppm in the ^11^B{^1^H} NMR spectra of all naphthyl-*o*-carborane compounds, which clearly confirmed the presence of the *o*-carborane cage.

The solid-state molecular structures of the four *o*-carboranyl compounds were also determined by X-ray single-crystallography (inset figures in Figure 1; detailed parameters, including selected bond lengths and angles, are provided in Appendix A). The crystal structures of the four compounds revealed planar naphthyl groups and icosahedral formation of *o*-carborane cages. Interestingly, bulky trimethylsilyl group-substituted compounds (**15CS** and **26CS**) showed a significantly perpendicular orientation between the naphthalene plane and the bonding axis of the C-C bond in the *o*-carborane cage (Ψ = 78° for **15CS** and 82° for **26CS**, Table 1), whereas the C-C bond axes of **15CH** and **26CH** were almost parallel to the naphthalene plane, as evidenced by the Ψ values (5.3° for **15CH** and 1.8° for **26CH**). Such a different geometry between silyl- and H-substituted compounds is a decisive factor in controlling their photophysical characteristics (vide infra).

### 3.2. Analysis of Photophysical Properties with Theoretical Calculation

The absorption and emissive properties of the naphthyl-bis-*o*-carboranyl compounds **15CH**, **26CH**, **15CS**, and **26CS** were investigated using UV/Vis absorption and PL spectroscopy, respectively (Figure 2 and Table 2). All compounds in THF exhibited major absorption bands centered at *λ*_abs_ = 278–301 nm, which were attributed to the vibronic structures undergoing spin-allowed π–π* transitions of the naphthalene moiety. These bands were also observed in the absorption spectrum of the mother-scaffold compound (naphthalene, *λ*_abs_ = 276 and 285 nm, Appendix A). However, the lowest absorption bands for the *o*-carboranyl compounds were relatively red-shifted (*λ*_abs_ = 322–330 nm) compared to the absorption band (*λ*_abs_ = 313 nm) of naphthalene because of the lowest unoccupied molecular orbital (LUMO) stabilization effect of *o*-carborane [67]. Furthermore, the low-energy absorption of these compounds includes tailing of the absorption bands at 350 nm. These features indicate that the bands can be attributed to the typical ICT transitions in the *o*-carborane cages. Time-dependent density functional theory (TD-DFT) calculations for the S_0_ state of the *o*-carboranyl compounds clearly confirmed these characteristics (vide infra).

The origin of the electronic transitions of the four naphthyl-*o*-carboranyl compounds was determined using TD-DFT calculations [60] (Figure 3). Each calculated structure was based on the X-ray crystal structure (Figure 1). The calculation results for the S_0_ state demonstrated that the major lowest-energy transitions were dominantly assignable to the highest occupied molecular orbital (HOMO) → LUMO (Figure 3). Although both the HOMO and LUMO of each naphthyl *o*-carborane compound were mainly localized on the naphthyl group (HOMO > 90% and LUMO > 84%, Appendix A), the LUMO levels were further distributed over the *o*-carborane cages (>12%). These results suggest that the lowest-energy electronic transition for the four *o*-carboranyl compounds originates from the naphthyl-centered π–π* LE transition with substantial ICT transitions between the naphthyl moiety and *o*-carboranyl cage.

The emissive properties of the four *o*-carboranyl compounds were examined using PL measurements under various conditions (Figure 2 and Table 2). Interestingly, the PL spectra of all the compounds in the THF solution at 298 K exhibited non-emissive characteristics. On the other hand, these compounds showed intense emissive patterns in rigid states (THF at 77 K and film), and the hydrogen-substituted *o*-carboranyl compounds **15CH** and **26CH** exhibited specific dual-emission patterns centered at *λ*_em_ = ~345 and ~480 nm, and trimethylsilyl-compounds **15CS** and **26CS** exhibited strong emission only in the region of *λ*_em_ = 470–500 nm. As a result, greenish emission was observed in the film states of all compounds under a hand-held UV lamp (Figure 2, inset figures). In particular, the emission of naphthalene was centered at *λ*_em_ = 322 and 336 nm (Appendix A), which proves that the high-energy emissions centered at *λ*_em_ = ~345 nm in the rigid states for **15CH** and **26CH** can be attributed to the LE transition of the naphthalene moiety. Moreover, the low-energy emissions of all the *o*-carboranyl compounds in the region over 450 nm were severely red-shifted compared with that of naphthalene, indicating that these corresponded to ICT involving *o*-carborane (vide infra). Such differences in the emission features between THF at 298 K and the rigid states typically originate from restricted structural fluctuations in the rigid state, and the elongation of the C-C bond in the *o*-carborane cage during the excitation process is known to diminish the efficiency of the ICT-based radiative decay mechanism [31,52,53,68,69,70]. Indeed, the calculated lengths of the C-C bonds for the *o*-carborane cages of all the compounds in the S_1_ state were significantly longer (ca. 2.40 Å, Table 1) than those in the S_0_ state (1.6–1.7 Å) and the experimentally measured values based on the solid-state crystal structures (ca. 1.7 Å).

The calculation results of the four *o*-carboranyl compounds in the S_1_-optimized structures revealed that the major transitions associated with low-energy emission involve a LUMO → HOMO transition (Figure 3). The LUMO and HOMO of the compounds were predominantly localized on the *o*-carborane cage (>76%, Appendix A) and the naphthyl group (>81%). These results strongly suggest that the low-energy emission around 450–550 nm observed in the rigid states is primarily manifested from a radiative decay process based on ICT between the *o*-carborane and naphthyl moieties.

To further investigate the origin of the greenish emission in the solid (film) state for these compounds, PL measurements were performed in a THF/distilled-water mixture (50 μM) (Figure 4 and Table 2). Intriguingly, the intensity of the low-energy emission for trimethylsilyl-substituted compounds **15CS** and **26CS** was dramatically enhanced in the region from 450 to 550 nm upon increasing the water fraction (*f*_w_). Consequently, the most aggregated state in THF/water (*f*_w_ = 90%) exhibited intense greenish emission (*λ*_em_ = 501 nm for **15CS** and 490 nm for **26CS**), which was significantly similar to that observed in the film state. These features are characteristic of aggregation-induced emission (AIE) phenomena. The structural rigidity could originate from the aggregation effect, resulting in an increase in the efficiency of the ICT-based radiative decay. Unfortunately, such an AIE phenomenon was not observed in the PL spectra of **15CH** and **26CH** THF/water mixtures.

### 3.3. Quantitative Comparison for ICT-Based Radiative Decay Efficiency

The absolute quantum efficiency (Φ_em_) and decay lifetime (τ_obs_) of each *o*-carboranyl compound in the film state were investigated to compare the efficiency of the radiative decay process quantitatively. Intriguingly, the Φ_em_ values of **15CS** and **26CS** were moderately estimated at 10.5% and 18.4%, respectively (Table 2), while those for **15CH** and **26CH** could not be measured owing to their inefficient radiative decay processes (<1%). Such differences in radiative efficiencies between the two series of naphthyl *o*-carboranyl compounds correlate with their structural geometry; in particular, the formation of the *o*-carborane cage appended to the plane of the aromatic group [40,41,47,48,49]. In general, the ICT transition on *o*-carboranyl luminophores can be intensified by increased delocalization through *exo*-π-interactions between the tangential *p*-orbital (σ*-character) on the substituted carbon atom of the *o*-carborane cage and the π-delocalized system of the appended aromatic group [40,41,71]. Consequently, the C-C bond axis of *o*-carborane becomes perpendicular to the aromatic plane to maintain the *exo*-π-interaction. Indeed, the calculated dihedral angles (Ψ) between the C-C bond axis and naphthyl plane in the optimized structures of the four compounds were orthogonally arranged according to the excited states (Ψ_S0_ = 30–85° → Ψ_S1_ = 82–90°, Table 1) for the σ*–π* conjugation between the carboranyl cages and naphthyl groups. However, the experimental Ψ values of **15CH** (5°, Table 1) and **26CH** (2°) from solid-state crystal structures indicated that the geometric relationship between the C-C bond axis and naphthyl planes was planar, whereas those of **15CS** (78°) and **26CS** (82°) were orthogonally maintained; the bulkiness of trimethylsilyl groups substituted to the *o*-carborane cages led to orthogonality. These structural features were the main reason why the radiative efficiencies of **15CH** and **26CH** in the solid state were lower than those of **15CS** and **26CS**. Furthermore, the restriction of *exo*-π-interaction for **15CH** and **26CH** generates the isolation of excitons on the naphthyl group [40], resulting in the hindrance of the ICT transition and the natural concentration of excitons to favor the LE transition from the naphthyl group. Consequently, **15CH** and **26CH** demonstrated strong LE-based emissions in the high-energy region (Figure 2 and Table 2), differing from those of **15CS** and **26CS**.

The τ_obs_ values for **15CS** and **26CS** in the film state were estimated to 3.7 and 6.4 ns, respectively (Table 2 and Appendix A), indicating distinctly fluorescent characteristics. A comparison of the radiative (*k*_r_; Table 2) and nonradiative (*k*_nr_) decay constants of both compounds, which were calculated by the Φ_em_ and τ_obs_ values, revealed a slight difference in the efficiency of the ICT-based radiative process between the two *o*-carboranyl compounds. The *k*_r_ values of both compounds were the same (2.9 × 10^7^ s^−1^, Table 2), whereas the *k*_nr_ value (2.4 × 10^8^ s^−1^) of **15CS** was approximately two times higher than that (1.3 × 10^8^ s^−1^) of **26CS**. These findings imply that the relatively low delocalization between the *o*-carborane cage and naphthyl group of **15CS** in the solid state leads to inefficiency of the ICT-based radiative decay process compared with **26CS**. Indeed, the experimental Ψ value (82°, Table 1) of **26CS** verified that this geometry was relatively closer to the orthogonal structure than that (78°) of **15CS**. Consequently, all the observed results verified that geometric formation around the *o*-carborane cage plays a significant role in controlling the efficiency of the ICT-based radiative decay mechanism on the aromatic group-appended *o*-carboranyl luminophores.

### 3.4. Theoretical Calculations of the Effect of Geometry around the O-carborane Cages

To elucidate the effect of the structural geometry around the *o*-carborane cages on the electronic transitions, the alternation of low-energy transitions assignable to ICT corresponding to the *o*-carborane (LUMO to HOMO in the S_1_-states, Figure 3) was calculated (B3LYP/6-31G(d,p) basis set) for **15CH** and **26CH** as the change in Ψ (the dihedral angle between the C-C bond axis of the *o*-carborane and naphthyl plane) from 0° (=360°) to 180° (Figure 5 and Appendix A). The orbital occupation of the LUMOs on the *o*-carborane cage for both **15CH** and **26CH** was maximized (> 95%) when the C-C axis was planar to the naphthyl plane (Ψ = 0° and 180°), whereas the localization of the HOMOs on the naphthyl groups did not change significantly (85–89% for **15CH** and 81–83% for **26CH**). In addition, the occupation of the LUMOs for both compounds gradually decreased as Ψ values had orthogonality (90°). These results indicate that the π-conjugated delocalization between the *o*-carborane and naphthyl group became blocked in accordance with being close to Ψ = 0° and 180°, and that the orbital distribution of each LUMO and HOMO was thoroughly isolated in the planar geometry, resulting in the inhibition of the ICT transition between the *o*-carborane and naphthyl groups. The oscillator strengths (*f*_calc_) computed for each transition of both compounds were maximized at Ψ = 90° and equaled zero at = 0° and 180°. These features strongly suggest that the orthogonality of the C–C bond axis to the plane of the appended naphthyl group significantly affects the ICT transitions of the *o*-carborane cage. The experimental and theoretical findings demonstrate that the structural geometry around the *o*-carborane cage plays a decisive role in enhancing the efficiency of the ICT-based radiative decay mechanism in the D–A-type *o*-carboranyl luminophores.

## 4. Conclusions

Herein, we prepared hydrogen (**15CH** and **26CH**) or trimethyl silyl groups (**15CS** and **26CS**) substituted with four naphthyl-based bis-*o*-carboranyl compounds and investigated the relationship between their molecular geometries and photophysical characteristics. Although these *o*-carboranyl compounds barely exhibited emissive patterns in solution at 298 K, intense greenish emission assignable to each LE- and ICT-based emission (for **15CH** and **26CH**) or ICT-based emission only (for **15CS** and **26CS**) were observed in rigid states (in solution at 77 K and in the film state). Intriguingly, the solid-state molecular structures of the four compounds revealed that the C-C bond axis of the *o*-carborane cage in the trimethysilyl group-substituted *o*-carboranyl compounds **15CS** and **26CS** was more orthogonal to the plane of the appended naphthyl group than those in the hydrogen-substituted *o*-carboranyl compounds **15CH** and **26CH**. These findings verify the efficient σ*–π* conjugation between the *o*-carborane and naphthyl groups in **15CS** and **26CS** via strong *exo*-π interaction. Such features result in a higher Φ_em_ for the emission of solid **15CS** and **26CS** than those of **15CH** and **26CH**. Furthermore, the most orthogonal geometry of **26CS** exhibits the highest Φ_em_ and lowest *k*_nr_ values for the ICT-based radiative decay process. Consequently, the results indicate that geometric formation around the *o*-carborane cage plays a decisive role in the efficiency of the ICT-based radiative mechanism in *o*-carboranyl organic luminophores.

## Figures and Tables

**Figure 1 molecules-27-06565-f001:**
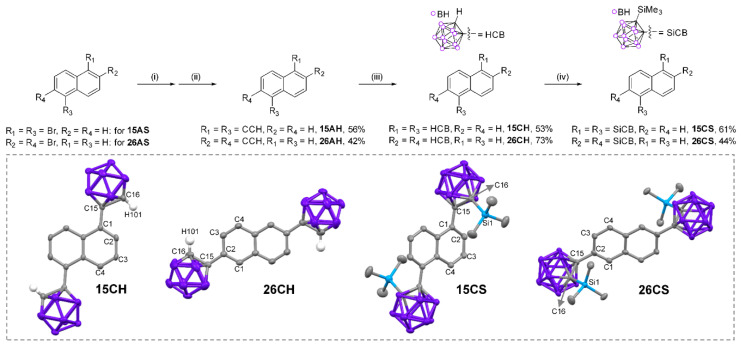
Synthetic procedures for the naphthyl-bis-*o*-carboranyl compounds. Reagents and conditions: **(i)** CuI, Pd(PPh_3_)_2_Cl_2_, ethynyltrimethylsilane, THF/NEt_3_, 80 °C, 24 h. **(ii)** K_2_CO_3_, methanol, 25 °C, 2 h. **(iii)** B_10_H_14_, N,N-dimethylaniline, toluene, 110 °C, 18 h. **(iv)** *n*-BuLi, TMSCl, THF, 25 °C, 2 h. Insets (dash-line box) show the X-ray crystal structures of **15CH**, **26CH**, **15CS**, and **26CS** (40% thermal ellipsoids), with H atoms on aromatic and aliphatic carbons omitted for clarity.

**Figure 2 molecules-27-06565-f002:**
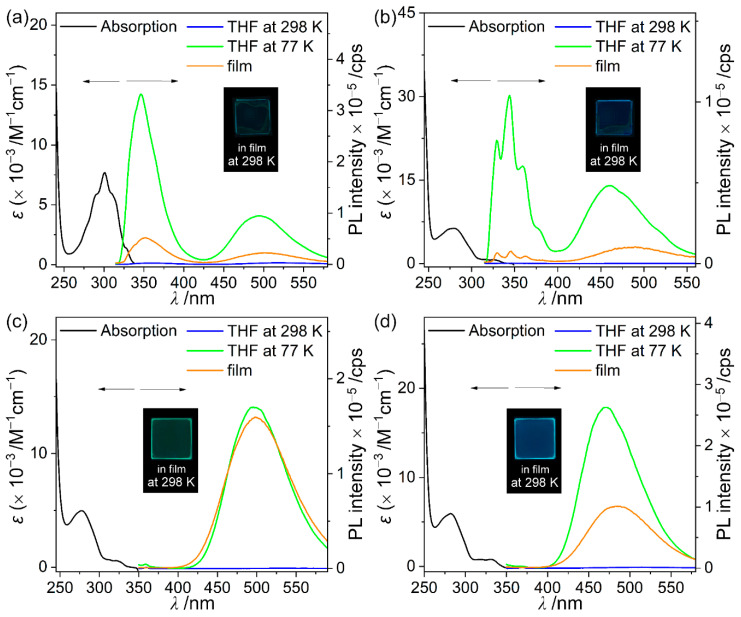
UV-vis absorption (left side) and PL spectra (right side) for (**a**) **15CH** (λ_ex_ = 305 nm), (**b**) **26CH** (λ_ex_ = 285 nm), (**c**) **15CS** (λ_ex_ = 329 nm), and (**d**) **26CS** (λ_ex_ = 333 nm). Black line: absorption spectra in THF (50 μM), blue line: PL spectra in THF (50 μM) at 298 K, green line: PL spectra in THF (50 μM) at 77 K, and orange line: PL spectra in film (5 wt% doped with PMMA) at 298 K. Inset figures show the emission color in the film state under irradiation by a hand-held UV lamp (λ_ex_ = 365 nm).

**Figure 3 molecules-27-06565-f003:**
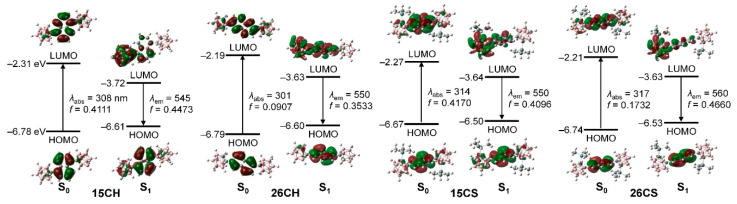
Frontier molecular orbitals for **15CH**, **26CH**, **15CS**, and **26CS** at their ground state (S_0_) and the first excited singlet state (S_1_) with their relative energies from DFT calculation (isovalue 0.04). The transition energy (in nm) was calculated using the TD-B3LYP method with 6-31G(d,p) basis sets.

**Figure 4 molecules-27-06565-f004:**
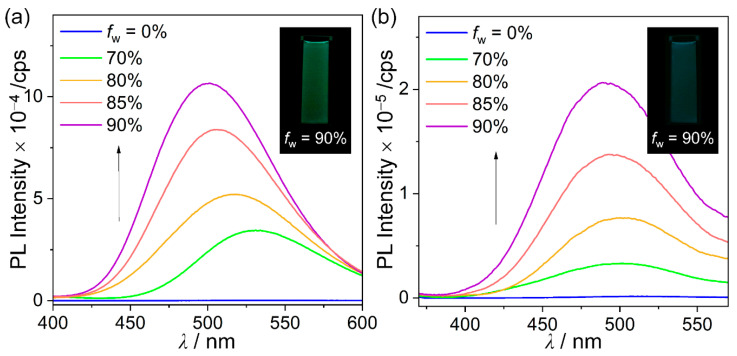
PL spectra of (**a**) **15CS** (*λ*_ex_ = 329 nm) and (**b**) **26CS** (*λ*_ex_ = 333 nm) in THF/distilled water mixtures (50 μM). Inset figures show the emission color in each state under irradiation by a hand-held UV lamp (*λ*_ex_ = 365 nm).

**Figure 5 molecules-27-06565-f005:**
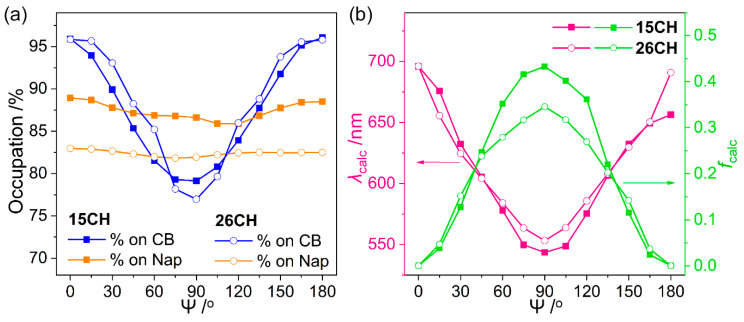
(**a**) Orbital contributions of the *o*-carborane cage (CB) and the naphthyl moiety (Nap) to the LUMO and HOMO, respectively, as a function of Ψ in S_1_-optimized structures for **15CH** and **26CH**. (**b**) Computed emission wavelengths (*λ*_calc_ in nm, pink lines) and oscillator strengths (*f*_calc_, green lines) for each low-energy transition (LUMO to HOMO).

**Table 1 molecules-27-06565-t001:** Dihedral angles between the naphthalene plane and the bonding axis of the C-C bond in the *o*-carborane cage (Ψ = C16–C15–C1(C2)–C2(C1) and bond lengths (C16–C15) of the C-C bond for **15CH**, **26CH**, **15CS**, and **26CS**.

	15CH			26CH			15CS			26CS		
	Exp.^1^	Calc.^2^		Exp.^1^	Calc.^2^		Exp.^1^	Calc.^2^		Exp.^1^	Calc.^2^	
		S_0_	S_1_		S_0_	S_1_		S_0_	S_1_		S_0_	S_1_
Ψ/°	5.3	43.4	89.3	1.8	30.9	89.2	78.0	74.9	82.5	82.0	85.1	89.7
C-C/Å	1.67	1.69	2.39	1.57	1.64	2.41	1.74	1.76	2.42	1.70	1.72	2.40

^1^ Experimental values from their X-ray crystal structures. ^2^ Calculated values from their ground (S_0_) and the first excited singlet state (S_1_) optimized structures.

**Table 2 molecules-27-06565-t002:** Photophysical data for naphthyl-bis-*o*-carboranyl compounds **15CH**, **26CH**, **15CS**, and **26CS**.

Compound	*λ*_abs_ ^1^/nm (ε × 10^−3^ M^−1^ cm^−1^)		*λ*_ex_ /nm	*λ*_em_/nm					
				THF ^2^		77 K ^1^	Film ^3^		*f*_w_ = 90% ^4^
**15CH**	301 (7.7), 328 (1.4)		305	-^8^		346, 495	349, 499		-^8^
**26CH**	279 (6.3), 326 (0.7)		285	-^8^		344, 460	345, 489		-^8^
**15CS**	278 (5.0), 322 (0.6)		329	-^8^		495	499		501
**26CS**	282 (5.9), 330 (0.8)		333	-^8^		471	485		490
**Compound**	**Φ_em_ ^3,5^**	**τ_obs_** ** ^3^ ** **/** **ns**			** *k* ** ** _r_ ** **^3,6^/× 10^7^ s** ** ^−1^ **			** *k* ** ** _nr_ ** **^3,7^/× 10^8^ s** ** ^−1^ **	
**15CH**	<0.01	-^7^			-			-	
**26CH**	<0.01	-^7^			-			-	
**15CS**	0.105	3.67			2.86			2.44	
**26CS**	0.184	6.40			2.88			1.28	

^1^ 50 μM in THF. ^2^ 50 μM, observed at 298 K. ^3^ Measured in the film state (5 wt% doped in PMMA). ^4^ 50 μM in THF/water (1/9, *v/v*). ^5^ Absolute PL quantum yield. ^6^ *k*_r_ = Φ_em_ /τ_obs_. ^7^ *k*_nr_ = *k*_r_(1/ Φ_em_−1). ^8^ Not observed due to weak emission.

## Data Availability

Data is contained within the article or Appendix A.

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
