# Peer review of "Relationship between the Molecular Geometry and the Radiative Efficiency in Naphthyl-Based Bis-Ortho-Carboranyl Luminophores"

_molecules, 2022, doi:10.3390/molecules27196565_

Round 1
Reviewer 1 Report
The manuscript describes photophysical properties of a series of naphthyl-bis-o-carboranyl compounds, in addition to their structural and theoretical studies. The research presented indicates a strong correlation between the molecular geometry and the efficiency of the ICT-based radiative decay mechanism. The Authors provided notable experimental and theoretical fundamental insights.
The manuscript deserves publication in Molecules. However, there are some minor points to be clarified before its publication:
1. Molecular packing of obtained compounds 15CH, 26CH, 15CS, and 26CS needs to be commented, especially that some of them show aggregation-induced emission (AIE) phenomena.
2. The information concerning functional, basis sets, and CPCM (“The ground (S0) and first excited (S1) states of the compounds were optimized at the B3LYP/6-31G(d,p) level of theory ……) should be moved from the Discussion to Experimental part.
Compounds 15CH, 26CH, 15CS, and 26CS should be rather assigned as non-emissive in THF at room temperature (instead of “the PL spectra of all the compounds in the THF solution at 298 K were barely visible”).
Author Response
REVIEWER REPORT(S):
On behalf of the author, I would like to thank the reviewer for reviewing our manuscript and for providing valuable suggestions. Our replies to the points you have raised are as follows and were considered in revising the manuscript.
Reply to reviewer #1
Comments to the Author
The manuscript describes photophysical properties of a series of naphthyl-bis-o-carboranyl compounds, in addition to their structural and theoretical studies. The research presented indicates a strong correlation between the molecular geometry and the efficiency of the ICT-based radiative decay mechanism. The Authors provided notable experimental and theoretical fundamental insights.
The manuscript deserves publication in Molecules. However, there are some minor points to be clarified before its publication:
Response: We thank the reviewer for the positive evaluation.
- Molecular packing of obtained compounds 15CH, 26CH, 15CS, and 26CS needs to be commented, especially that some of them show aggregation-induced emission (AIE) phenomena.
Response: Thank you for this comment. We totally agree with your comment that molecular packing system is generally concerned with the AIE effect. However, we would like to mention that these AIE phenomena for the two o-carboranyl compounds, 15CS and 26CS, are not derived from intermolecular interactions (such as π‒π stacking, hydrogen bonding and so on) but, originated from the structural rigidity though molecular aggreation, which can enhance the efficiency of the ICT-based radiative decay. This is general effect in the aromatic group appended o-carboranyl luminophores (corresponding literatures: Angew. Chem. Int. Ed. 2015, 54, 5084; Eur. J. Inorg. Chem. 2021, 46, 4875; Inorg. Chem. Front. 2022, 9, 501; J. Mater. Chem. C 2017, 5, 10047). As specified, we have revised the description in the manuscript (highlighted in yellow on page 7, line 314‒315).
- The information concerning functional, basis sets, and CPCM (“The ground (S0) and first excited (S1) states of the compounds were optimized at the B3LYP/6-31G(d,p) level of theory ……) should be moved from the Discussion to Experimental part.
Response: As mentioned in the comment, we removed the corresponding description in the Discussion section and presented the computational details in Computational Calculation Studies of Experimental section.
Compounds 15CH, 26CH, 15CS, and 26CS should be rather assigned as non-emissive in THF at room temperature (instead of “the PL spectra of all the compounds in the THF solution at 298 K were barely visible”).
Response: As mentioned in the comment, we revised the corresponding description in the manuscript (highlighted in yellow on page 7, line 281‒282).

Reviewer 2 Report
The paper under review, Relationship Between the Molecular Geometry and the Radiative Efficiency in Naphthyl-based Bis-ortho-Carboranyl Luminophores, deals with synthesis, structural characterization, and luminescent properties of four newly synthesized compounds.
The experimental work seems carefully done although the x-ray structure of 15CH has a R1 and GOF values that are suspicious of a not so good structure (R1) and deficient weighting scheme (GOF). However, it's good enough for the proposed use in the discussion of this paper.
My major concern goes to the computational calculations:
- The chosen functional is not appropriate for photophysical calculations due to the wrong behavior decay of the charge density. The corresponding range separated CAMB3LYP or better yet the LR-BPBE(w=0.2). With the chosen functional I would expect a blue shift of all the computed bands between 0.7 and 1 eV.
- THE MAJOR PROBLEM that must be addressed before acceptance is the calculations in S1 where a 240 pm C-C distance was found. This distance is not even a non-covalent interaction. I think the alleged ICT calculated transition is that of the naphthalene anion with a caged cation nearby. The radical anion is always red shifted (by an amount similar to that obtained by the authors) when compared to the neutral species. Even if chemically possible at that distance the S1 state would become instantaneously a triplet state. Being dissociative the experimental essay would be destructive under continuous irradiation.
The authors should check what’s wrong. Here is my suggestion:
If the S1 state is the LE state, the S2 state should be the ICT. During the C-C stretching the S1 and S2 states cross. At that point of S1/S2 degeneracy the program was unable to figure out the right state to optimize. Probably the authors have to perform an IRC quick scan at fixed C-C geometries to detect the states intersection and figure out manually which state to optimize after that point. No guarantee that would work by I leave it as a suggestion.
Author Response
REVIEWER REPORT(S):
On behalf of the author, I would like to thank the reviewer for reviewing our manuscript and for providing valuable suggestions. Our replies to the points you have raised are as follows and were considered in revising the manuscript.
Reply to reviewer #2
Comments to the Author
The paper under review, Relationship Between the Molecular Geometry and the Radiative Efficiency in Naphthyl-based Bis-ortho-Carboranyl Luminophores, deals with synthesis, structural characterization, and luminescent properties of four newly synthesized compounds. The experimental work seems carefully done although the x-ray structure of 15CH has a R1 and GOF values that are suspicious of a not so good structure (R1) and deficient weighting scheme (GOF). However, it's good enough for the proposed use in the discussion of this paper.
Response: We thank the reviewer for the positive evaluation.
My major concern goes to the computational calculations:
-The chosen functional is not appropriate for photophysical calculations due to the wrong behavior decay of the charge density. The corresponding range separated CAMB3LYP or better yet the LR-BPBE(w=0.2). With the chosen functional I would expect a blue shift of all the computed bands between 0.7 and 1 eV.
Response: Thank you for your suggestion. In fact, we are not expert for the computational calculations and would like to just obtain the electronic information from simple DFT calculation results via Gaussian program. The basis function B3LYP is the easiest way we can approach and furthermore, it showed the best similar results with the given experimental information. There are already lots of reports for the computational results based on B3LYP functional to elucidate the origin of electronic transition on the o-carboranyl luminophores. We believe that this method looks not appropriate for severe investigation, but can sufficiently give the hint for the electronic transition of the o-carboranyl compounds.
- THE MAJOR PROBLEM that must be addressed before acceptance is the calculations in S1 where a 240 pm C-C distance was found. This distance is not even a non-covalent interaction. I think the alleged ICT calculated transition is that of the naphthalene anion with a caged cation nearby. The radical anion is always red shifted (by an amount similar to that obtained by the authors) when compared to the neutral species. Even if chemically possible at that distance the S1 state would become instantaneously a triplet state. Being dissociative the experimental essay would be destructive under continuous irradiation. The authors should check what’s wrong. Here is my suggestion: If the S1 state is the LE state, the S2 state should be the ICT. During the C-C stretching the S1 and S2 states cross. At that point of S1/S2 degeneracy the program was unable to figure out the right state to optimize. Probably the authors have to perform an IRC quick scan at fixed C-C geometries to detect the states intersection and figure out manually which state to optimize after that point. No guarantee that would work by I leave it as a suggestion.
Response: We highly appreciate your comments and suggestion. First of all, the longer the C−C bond length in the o-carborane cage become, the more concentrated the electron (charge) density is (the experimental longest bond length = 2.022 Å; Inorg. Chem. 2002, 41, 3347. And also, diamino-o-carborane showed the 1.93 Å; Angew. Chem. Int. Ed. 2019, 58, 1397.). In addition, many previous reports had exhibited the similar results for the bond lengths of the o-carborane cages in excited state (corresponding literatures: Angew. Chem. Int. Ed. 2012, 51, 2677; Org. Lett. 2016, 18, 4064; Eur. J. Inorg. Chem. 2021, 46, 4875; Inorg. Chem. Front. 2022, 9, 501; Chem. Eur. J. 2020, 26, 548; J. Mater. Chem. C, 2021, 9, 9874). Indeed, we do not also believe that the C‒C bond is actually over 2.4 Å in excited state. However, these theoretical results have given the insight for the geometric feature during the excitation and relaxation process; the C‒C bonds for the compounds are severely fluctuated in the process via the focused charge density to the o-carborane cages in excited state.
In addition, we regretfully disagree with the reviewer’s comments that the S1 state is the LE state, the S2 state should be the ICT. In general, the LE-based electronic transition is higher than the ICT-based transition in the aromatic appended o-carboranyl luminophores (corresponding literatures: Chem. Eur. J. 2014, 20, 9940; Angew. Chem. Int. Ed. 2015, 54, 5084; Inorg. Chem. Front. 2022, 9, 501; Chem. Eur. J. 2020, 26, 548; J. Mater. Chem. C, 2020, 8, 16896; Mater. Chem. Front. 2022, 6, 783). Such the previous reports showed that the ICT transition induced the red-shifted absorption and emissive characteristics in the photophysical properties of the o-carboranyl luminophores.
We totally agree with the reviewer’s comment that it looks a little strange to describe the electronic transitions for the frontier orbital levels (including HOMO and LUMO) in the S1-optimized structures. In addition, it is rather difficult to find clear evidence for experimentally verifying the transition process. However, the one thing we would like to mention is that the molecular geometry changes during the excitation and relaxation process, and as a result, the electronic transitions in the S0 (ground state)-optimized structure cannot give information relating to the relaxation process (a radiative and non-radiative decay process) because the molecular structure in the excited state (S1-state) is apparently different from that in the ground state. Thereby, we believe that analysis of the electronic transitions in the S1-optimized structure should be separately performed to elucidate the origin of the emissive characteristics, and we believe that this represents the best method reported thus far. Furthermore, these calculation results were relatively well-matched with the experimental results, in particular the emissive features. We expected that these calculation techniques, which can simultaneously investigate the electronic transitions in accordance with the molecular dynamics between the ground and excited states, will be developed and improved in the near future.
Subsequently, we obtained the energy levels and orbital distribution of the frontier orbitals (HOMOs and LUMOs) and their electronic transitions after excited (S1) geometry optimization using the time-dependent (TD) calculation. In addition, we individually defined the HOMO/LUMO orbitals as the singlet spin-state energy level, particularly the two frontier orbitals occupied with a +1/2 spin-state electron are the HOMO and LUMO levels, respectively. Accordingly, we considered a system wherein a single electron was excited from the HOMO to LUMO in each S1-optimized geometry for the o-carboranyl compounds.

Round 2
Reviewer 2 Report
In my opinion, the authors reply to reviewers adresses in the correct direction the points raised in the reviewing process. The calculations are not in the core of the paper and less than state of the art calculations could be used in the discussion of results. However, I would feel more confortable if a statment of caution pointing to that was included in the manuscript. Future readers have no acess to reviewer comments and answers.
Author Response
Reply to reviewer #2
Comments to the Author
In my opinion, the authors reply to reviewers adresses in the correct direction the points raised in the reviewing process. The calculations are not in the core of the paper and less than state of the art calculations could be used in the discussion of results. However, I would feel more confortable if a statment of caution pointing to that was included in the manuscript. Future readers have no acess to reviewer comments and answers.
Response: We thank the reviewer for the positive evaluation. As the reviewer suggested, we added the ‘Caution’ description for the calculation method in the Computational Calculation Studies of Experimental section (highlighted in yellow, page 5).